# Respiratory Syncytial Virus Infection Induces Chromatin Remodeling to Activate Growth Factor and Extracellular Matrix Secretion Pathways

**DOI:** 10.3390/v12080804

**Published:** 2020-07-26

**Authors:** Xiaofang Xu, Dianhua Qiao, Morgan Mann, Roberto P. Garofalo, Allan R. Brasier

**Affiliations:** 1Department of Medicine, University of Wisconsin-Madison School of Medicine and Public Health (SMPH), Madison, WI 53705, USA; xxu@wisc.edu (X.X.); dqiao@wisc.edu (D.Q.); mwmann@wisc.edu (M.M.); 2Department of Pediatrics, University of Texas Medical Branch, Galveston, TX 77555, USA; rpgarofa@utmb.edu; 3Institute for Human Infections and Immunity, University of Texas Medical Branch, Galveston, TX 77555, USA; 4Institute for Clinical and Translational Research (ICTR), University of Wisconsin-Madison, Madison, WI 53705, USA

**Keywords:** assay for transposase-accessible next generation sequencing (ATAC-Seq), chromatin remodeling, innate response, hexosamine biosynthetic pathway

## Abstract

Lower respiratory tract infection (LRTI) with respiratory syncytial virus (RSV) is associated with reduced lung function through unclear mechanisms. In this study, we test the hypothesis that RSV infection induces genomic reprogramming of extracellular matrix remodeling pathways. For this purpose, we sought to identify transcriptionally active open chromatin domains using assay for transposase-accessible-next generation sequencing (ATAC-Seq) in highly differentiated lower airway epithelial cells. High confidence nucleosome-free regions were those predicted independently using two peak-calling algorithms. In uninfected cells, ~12,650 high-confidence open chromatin regions were identified. These mapped to ~8700 gene bodies, whose genes functionally controlled organelle synthesis and Th2 pathways (IL6, TSLP). These latter cytokines are preferentially secreted by RSV-infected bronchiolar cells and linked to mucous production, obstruction, and atopy. By contrast, in RSV infection, we identify ~1700 high confidence open chromatin domains formed in 1120 genes, primarily in introns. These induced chromatin modifications are associated with complex gene expression profiles controlling tyrosine kinase growth factor signaling and extracellular matrix (ECM) secretory pathways. Of these, RSV induces formation of nucleosome-free regions on *TGFB1/JUNB//FN1/MMP9* genes and the rate limiting enzyme in the hexosamine biosynthetic pathway (HBP), Glutamine-Fructose-6-Phosphate Transaminase 2 (*GFPT2*). RSV-induced open chromatin domains are highly enriched in AP1 binding motifs and overlap experimentally determined JUN peaks in GEO ChIP-Seq data sets. Our results provide a topographical map of chromatin accessibility and suggest a growth factor and AP1-dependent mechanism for upregulation of the HBP and ECM remodeling in lower epithelial cells that may be linked to long-term airway remodeling.

## 1. Introduction

Respiratory syncytial virus (RSV) is single-stranded, negative-sense member of the *Orthopneumovirus* genus of the Pneuomoviridae family that represents a major cause of acute pulmonary disease in children, adults with chronic airway disease and the hospitalized elderly. Without an effective vaccine, virtually all children are infected by the age of two [1]. A subgroup of those infected, estimated to number ~2.1 million children under five years of age, require medical attention in the US annually [2]. Consequently, RSV represents the most common cause of pediatric hospitalization in children less than five years of age [3]. In immunologically naïve infants, a subset will develop severe lower respiratory tract infections (LRTIs), such as bronchiolitis and pneumonia. Through incompletely understood mechanisms, LRTIs are linked to recurrent virus induced wheezing [4], atopy [5], and decreased lung function [6]. 

RSV replication is restricted to the airway mucosa where the virus primarily replicates in the epithelium. The nature, timing, and magnitude of inducible epithelial cytokine responses shape the initial anti-viral responses to acute infection [7] and influence evolution of the adaptive immune response [8]. In epithelial cells, RSV RNA replication occurs in cytoplasmic inclusion bodies. This event is primarily sensed by a cytoplasmic ATP-dependent RNA helicase known as retinoic acid inducible gene (RIG)-I [9]. Activated RIG-I binds to the mitochondrial antiviral signaling protein (MAVS) via homotypic interactions of caspase activation and recruitment domains (CARD) on both proteins to activate formation of the inflammasome on the mitochondrial surface. The inflammasome is a signaling receptor complex containing kinases and ubiquitin ligases that are rate-limiting in activating regulatory transcription factors including interferon regulatory factor (IRF)3, nuclear factor-κB (NFκB), Jun proto-oncogene (JUN) and others. Consequently, inflammatory networks of genes are expressed responsible for neutrophilic inflammation and anti-viral immunity [10,11]. 

In human RSV challenge studies, although only mild disease is observed clinically in adult volunteers, RSV spreads to the lower airway where it produces prolonged mucosal inflammation [12]. In contrast to epithelial cells from the conducting airways, lower airway bronchiolar cells play a key role in the pathogenesis of RSV-induced inflammation and obstruction [13]. In mice, small airway epithelial cells of the distal bronchiole play a central role in RSV-induced neutrophilic inflammation, mucous production, and obstruction [7]. Specifically, these bronchiolar cells synthesize Th2-polarizing, mucogenic, and profibrotic cytokines to a greater degree than airway cells of the conducting airways [10,11]. Cell-type selective gene disruption of the innate pathway in these lower nonciliated bronchiolar cells interferes with airway neutrophilia, cytokine responses, and airway obstruction [7]. These findings provide direct evidence for the presence of distinct regional, cell type-specific responses to RSV infection. The mechanisms underlying cell-type differences in response are largely unexplored. 

Although it is clear that epigenetic regulation plays a key role in the complex epithelial response to RSV infection, the phenomenon has not been studied systematically. The location and characteristics of constitutively active open chromatin domains and how these are modified by RSV infection are not known. In this study, we apply high resolution assay for transposase-accessible chromatin (ATAC)-next generation sequencing (Seq) to advance our understanding of epigenetic regulation in RSV infection. ATAC-Seq was performed in resting (control) and RSV-infected human small airway epithelial cells (hSAECs). Using two independent peak-calling algorithms to identify high-confidence nucleosome-free regions, we observe resting hSAECs have nucleosome-free domains in promoters controlling cell cycle regulation and protein glycosylation. Importantly, T helper 2 (Th2) cytokines preferentially expressed by RSV infected lower airway cells, including interleukin 6 (IL6), thymic stromal lymphopoietin (TSLP), CCL-20, and macrophage inducible protein (MIP3α) are found in open chromatin domains. We observe that RSV induces nucleosome repositioning of distinct genic sequences associated with complex mRNA expression profiles. Functional pathway analysis of RSV-induced open domains on gene bodies indicates enhanced chromatin accessibility underlies TGFβ-induced extracellular matrix (ECM) formation and remodeling pathways. These data provide the first direct evidence that RSV produces epigenetic reprogramming controlling tissue remodeling. 

## 2. Materials and Methods 

### 2.1. Cell Culture and Virus Isolation

Human small airway epithelial cells (hSAECs) were immortalized using human Telomerase/CDK4 as previously described [14,15]. hSAECs were grown in SAGM small airway epithelial cell growth medium (Lonza, Walkersville, MD, USA) in a humidified atmosphere of 5% CO_2_. The human RSV long strain was grown in Hep-2 cells and prepared as described [7,16]. The viral titer of purified RSV pools was varied from 8 to 9 log PFU/mL, determined by a methylcellulose plaque assay [17]. Viral pools were aliquoted, quick-frozen on dry ice-ethanol, and stored at −70 °C until they were used.

### 2.2. Next Generation RNA Sequencing

Total RNA was isolated using RNeasy with on-column DNase I digestion (Qiagen, Germantown, MD, USA) and integrity verified spectroscopically and by electrophoresis prior to sequencing. *n* = 4 independent samples were analyzed at time 0, 16 h, and 24 h of RSV infection (MOI = 1.0). Samples were bar-coded and subjected to Illumina HiSeq 2000 (San Diego, CA, USA) paired end sequencing in the UW BioTech Center. The trimming software skewer was used to process raw fastq files and QC statistics computed. RNA sequences corresponding to a Phred score of 30 (99.9% base call accuracy) were used for alignment. The trimmed paired-end reads were aligned against human genome hg19 using the program, Spliced Transcripts Alignments to a Reference (STAR, version 2.7.5a). Mapped paired-end reads for both genes and transcripts (isoforms) were counted in each sample using RNASeq by Expectation Maximization (RSEM). Hierarchial Clustering (HC) was performed to identify patterns in gene expression in the pHeatMap program in R (version 1.0.12).

### 2.3. Assay for Transposase-Accessible Chromatin Sequencing (ATAC-Seq)

ATAC-seq was performed in two biological replicates per treatment. Briefly, 50,000 cells (>95% viability) were trypsinized and washed. Cell pellets were re-suspended in 300 µL of cold lysis buffer (10 mM Tris-HCl, pH7.4, 10 mM NaCl, 3 mM MgCl_2_, 0.1% IGEPAL CA-630) and incubated on ice for 10 min. Nuclei were pelleted, re-suspended in 50 µL of transposition mixture (25 µL 2× TD buffer (Illumina, Cat. No.: 15027866, San Diego, CA, USA), 2.5 µL TDE1 (Illumina Nextera Tn5 transposase, Cat. No.: 15027865), 22.5 µL nuclease-free water) and incubate for 30 min at 37 °C water bath with gentle mixing.

After purification (Qiagen MinElute PCR purification kit, Qiagen, Germantown, MD, USA, Cat. No.: 28204), eluted DNA was pre-amplified for 5 cycles in PCR mix (30 µL NEB Next High-Fidelity 2x master mix (NEB, Cat. No.: M054 1S), 10 µL primer mix (Illumina, NEXtera DNA UD indexes, San Diego, CA, USA; Cat. No.: 20026121), 20 µL purified DNA for 5 cycles. An additional 3 amplification cycles were added based on QPCR amplification. PCR reactions were then purified. Libraries were analyzed on a bioanalyzer and sequenced on a NovaSeq 6000 (Illumina, San Diego, CA, USA) with 50 million reads per sample.

### 2.4. ATAC-seq Data Analysis

Fastq files were analyzed for read coverage, PCR duplication and number of mapped reads determined using FastQC (version 0.11.8). Adapter removal was using TrimGalore [18]. Alignment was performed using RSubread package (version 3.11, [19]) to the GRCh38.p13 (hg38) genome assembly (NCBI). Peak calling was performed using Genrich for ATAC-Seq (c 2018, [20]), using commands to remove mitochondrial sequences, remove PCR duplicate reads and mask blacklisted sequences. Separate analysis was using hidden markov modeleR (HMMR, version 1.2.4) optimized for ATAC-Seq [21]. Principal components analyses (PCA) and correlation analyses were conducted in the DiffBind package [22]. Statistical analysis was performed using DESEQ2 in the DiffBind package (version 2.16.0), accepting an adjusted *p* value of <0.05 as significant. Annotation of high confidence peaks was performed using ChIPSeeker in the R environment (version 1.24.0, [23]). Functional annotation was performed in ReactomePA and KEGG using DOSE (version 3.8.2) and ReactomePA packages in R [24]. Motif enrichment of peaks for known motifs was performed using HOMER (version 4.11, [25]). This algorithm uses a binominal distribution accepting motifs that occur at 0.1% false discovery rate (FDR) or less.

ATAC seq data is deposited in Genome Expression Omnibus (GEO) and will be made publicly available upon publication.

## 3. Results

We examined the resting and inducible chromatin landscape in human small airway epithelial cells (hSAECs). hSAECs are derived from *Scgb1a1+*-expressing progenitor cells, representing the distal bronchiole phenotype associated with pathogenicity of RSV pneumonia in vivo [7]. These cells also maintain a highly differentiated epithelial phenotype without entering senescence in cell culture [15], making this model robust for profiling chromatin landscape during cell state transitions [14,26]. Our previous work shows that hSAECs support RSV replication, and activate innate signaling in a dose and MOI-dependent manner [13,27,28]. At a multiplicity of infection (MOI) of 1, hSAECs express a time-dependent activation of chemokines, cytokines and anti-viral IFNs that peak at 24 h without evidence of cell death [11]. Finally, hSAECs exhibit viral induced genomic and proteomic expression patterns that are highly correlated with those of primary small airway epithelial cells, making this system informative [11,14].

Uninfected (control) and RSV infected hSAECs (MOI = 1, 24 h) were subjected to ATAC-Seq and paired end reads were obtained on a NovaSeq 6000. An average of 55 M reads were obtained; 88% of the reads were mapped to the hg38 genome (Appendix A). After adapter removal, the fragment lengths showed a typical size distribution of 100 nt with an asymmetric tail characteristic of mono- and di-nucleosomal fragmentation (Appendix A). To examine the distribution of fragments, library complexity patterns were analyzed. This analysis indicated that complexity was similar among replicates Appendix A).

To examine whether the ATAC-Seq peaks were reproducible and similar to its replicate, correlation metrics were calculated and subjected to hierarchical clustering. We noted that each treatment condition correlated with its replicate for both control and RSV-infected samples (Figure 1A). Principal Component Analysis (PCA) was also performed to determine sample variability and similarity between replicates. We observed that the control and RSV-treated samples were widely separated in the PCA first dimension. This dimension accounted for 88% of the variability of the sample, indicating that the data represented a robust effect of RSV infection (Figure 1B). We concluded from this quality analysis that the ATAC-Seq data was reproducible and informative of the RSV effect.

### 3.1. Open Chromatin Domains in Control Cells—“Occupancy Analysis”

Because the Tn5 transposase cleaves nucleosome-free DNA, the ATAC-Seq reads of the uninfected (control) hSAECs were first analyzed to identify constitutively open regions. ATAC-Seq reads from control cells were analyzed to identify the landscape and characteristics of these accessible regions. To this end, we first evaluated the performance of different bioinformatics tools to distinguish peaks from noise. Recent developments in peak calling algorithms have resulted in the advancement of several tools that use different approaches and assumptions. For this work, we applied two peak-calling algorithms to identify high-confidence peaks. The first algorithm, Genrich, is adapted from the Galaxy Project for analysis of paired end ATAC-Seq data, creating intervals where Tn5 cuts occur, removing mitochondrial sequences and PCR duplicates. The second algorithm, Hidden Markov Modeling for ATAC (HMMRATAC), is a machine learning method that splits the ATAC-seq dataset into nucleosome-free and nucleosome-enriched signals, learns the unique chromatin structure around accessible regions, and then predicts accessible regions across the entire genome [21].

Genrich and HMMRATAC independently identified ~12,650 chromatin peaks that were considered high confidence peaks. The overlap of these two predictions were compared by examining those that mapped to annotated gene bodies. Peaks were identified in ~8700 gene bodies shown in the Venn diagram in Appendix A. To understand the locations of ATAC-Seq peaks on gene bodies in more detail, their distribution were mapped to canonical gene locations, including the promoter, 5′ untranslated tracts (UTRs), gene bodies (exon/intron), 3′ UTR, and distal intergenic sequences. These are represented as annotation plot and an “upset” plot (Figure 2A). The upset plot is a condensed method for examining the co-localization of multiple peaks on a gene body. From this analysis, the majority of peaks (*n* = ~3500) were distributed in the Promoter, 5′ UTR, Exon, Intron, and 3′ UTR (Figure 2A). The second most common location of ATAC-Seq peaks (*n* = ~2800) were located only within introns of genes. These data indicated that open chromatin domains in control cells were primarily located within gene bodies.

We next examined the distribution of peaks across the chromosomes (Chrs). In this analysis, ATAC-Seq peaks were mapped to individual chromosomes, where each chromosome is represented as a linear entity scaled to its relative length. We noted ATAC-Seq peaks were distributed across the chromosomes as well as both sex chromosomes (Figure 2B). We noted dense clusters of Tn5 cleavage on chromosomes Chrs-5 and -9, with regions of Chrs-13–15 and -22 being devoid of peaks. To further examine the location of ATAC-Seq peaks, the peaks were represented as a tagged heat map, where the length of the sequenced fragment was mapped relative to the annotated transcription start site (TSS, Figure 2C). We noted a symmetric distribution of fragments over the TSS, centered over the origin (0, Figure 2C). As confirmation, we plotted frequency of read counts relative to the origin of the TSS. A symmetrical peak was observed centered over the TSS (Figure 2D). These data are consistent with the gene body annotation (c.f. Figure 2A,C,D). We interpret these data to indicate that the regions of open chromatin domains in control cells are located primarily in the promoters of genes centered on TSSs.

### 3.2. Nucleosome-Free Regions Are Associated with Organelle Function and Cell Cycle Regulation

To determine the functions of the genes within open chromatin domains, high confidence ATAC-Seq peaks found within 3000 nt of the TSS of a known gene (over two-thirds were <1 kB of the TSS) were analyzed by Reactome pathway analysis. Reactome pathways are a manually curated annotation pathway database, where pathway enrichment is identified by “Gene ratio”, where the number of genes is expressed as a fraction of the total number of genes in that pathway [29]. Significance is determined by an adjusted *p*-value (padjust). We observed that functional pathways associated with genes in open chromatin domains were significantly associated with organelle biogenesis, *N*-linked protein glycosylation, and cell cycle regulation (Figure 3).

### 3.3. Open Chromatin Domains Are Associated with Active Promoter Marks

To further confirm that open chromatin domains are enriched in active promoters, ATAC-Seq peaks were compared with publicly accessible ChIP-Seq data sets in GEO. We observed that the frequency of read counts in control hSAECs significantly overlapped with experimentally determined RNA polymerase II (Pol II), and histone-3 (H3) Lys (K27) acetylation (Ac) peaks, but were not enriched in repressed gene marks, such as H3K27me3 (Figure 4A–C). We note that the H3K27Ac peaks were broader than the narrow peaks produced by RNA Pol II and other sequence-specific transcription factors. The H3K27Ac mark is significant because this modification is a characteristic of transcriptional elongation occurring via the super elongation complex [27]. These data suggest open chromatin domains in resting cells are enriched for transcriptionally active promoters engaging RNA Pol II and associated with histone marks associated with active transcriptional elongation.

Our previous work comparing differential protein expression between hSAECs and bronchial cells derived from conducting airways identified a cluster of Th2-regulating and mucogenic cytokines to be preferentially expressed by hSAECs [11]. This cluster included including IL6, TSLP, CCL-20, and MIP-3α. We therefore examined Tn5 cleavage patterns on these promoters to see if these genes are contained within open chromatin domains in uninfected cells. Substantial cleavage patterns corresponding to the promoters, introns, and gene bodies of IL6 and TSLP were observed (Figure 4D,E). These findings extend our understanding on the mechanisms for regulation of Th2 cytokine production by lower airway epithelial cells.

### 3.4. Overlap with Functional Enhancer Sequences

Collectively, the gene annotation, tagged matrix analysis, peak enrichment centered on TSSs, and co-occurrence of RNA Pol II ChIP-Seq peaks suggested the ATAC-Seq cleavage sites are enriched in proximal promoter sequences. Nevertheless, ~20% of ATAC-Seq annotations are located in intergenic/distal intergenic regions (Figure 2A) and the H3K27Ac mark also has been associated with active enhancers, complicating this interpretation. To further examine whether enhancer sequences are represented, we extracted cell-type-specific enhancers from an annotated database. It is known that active enhancers express bidirectional enhancer RNAs (eRNAs); these transcripts can be identified using global run-on sequencing (GroSeq, [30]). We obtained active enhancer sequences from a human alveolar (A549) cell from an annotated database of bidirectional GroSeq studies [31]. Enhancer sequences were then compared to high-confidence chromatin domains. We observed that there was little overlap of active enhancers with the high-confidence ATAC-Seq peaks. Only seven enhancers were present in both, representing <0.05% of the peaks (Figure 5). Collectively, we interpret these data to indicate that the open chromatin domains in resting hSAECs correspond to proximal promoters of actively transcribed genes.

### 3.5. Promoter-Enriched Open Chromatin Domains Are Enriched in Basic Domain-Leucine Zipper (bZIP) Binding Motifs

A combinatorial code of cis regulatory elements on gene promoters confers cell-type and virus-inducible gene expression patterns by binding cognate transcription factors regulated by distinct signal transduction pathways. To identify if there are similar patterns of cis regulatory elements in open promoters, we systematically searched for enrichment of known motifs. For this purpose, we used HOMER, an algorithm that identifies motif enrichment relative to random DNA sequence matched for similar guanosine-cytosince (GC) content [24]. Over 150 highly significant motifs were identified, including members of the activator protein 1 (AP-1) complex, nuclear transcription factor Y (NFY), Kruppel Zinc Finger (KLF), specificity protein 1 (SP1) transcription factor binding sites, and others (Table 1). Interestingly, these factors are found at a significant fraction of the chromatin regions, with AP-1/ATF binding sites in 15% of regions, KLF in 16%, and SP2 in 22% of regions (Table 1).

### 3.6. RSV Inducible Changes in Chromatin Accessibility

To identify virus-inducible changes in chromatin accessibility, ATAC-Seq peaks were analyzed for differential expression. ATAC-Seq peaks in RSV-infected cells were compared to those from uninfected cells using a DESEQ2 contrast in DIFFBIND (comparing count abundance). 1700 differentially cleaved sites were identified in both peak-calling algorithms. These differentially expressed peaks mapped to 1120 gene bodies. In contrast to the gene-body annotations seen in the occupancy analysis of resting hSAECs, RSV-induced ATAC-Seq peaks were primarily found on introns. As shown in the gene annotation “upset plot”, ~700 ATAC-Seq peaks were found in a gene intron only, with 500 peaks located in a distal intergenic sequence (Figure 6A). These locations were distributed over all chromosomes, with more pronounced clustering on Chr 5 and Chr 9, and in a similar pattern to the peaks found in uninfected hSAECs (Figure 6B).

The RSV-inducible peaks were then mapped to annotated transcriptional start sites (TSSs) from human genome 38. We noted the RSV-induced ATAC-Seq peaks centered on the TSSs of annotated genes (Figure 6C). However, in contrast to the smooth tails of the constitutive ATAC-Seq peaks, the flanking sequences of the RSV-induced ATAC-Seq peaks showed periodicity (compare Figure 6C to Figure 2D). Of note, a larger peak 3′ to the TSS was identified in the RSV-inducible Data set. This asymmetric distribution is consistent with the intronic enrichment of the RSV-induced chromatin domains noted in Figure 6A.

Transcription factor binding enrichment of the RSV-inducible ATAC-Seq peaks was next examined using HOMER to identify enriched cis-regulatory motifs. Like the results of the analysis of ATAC-Seq from uninfected hSAECs, HOMER identified a highly significant enrichment of ATF sequences (adj *p* value of 1 × 10^−187^) in 19% of the RSV-induced nucleosome-free domains. But, in contrast, a different pattern of co-occuring motifs was identified. These sequences included the p53-related TP63, Transcriptional Enhancer Factor Domain (TEAD)3, forkhead (FOXA1), nuclear factor (NF)1 and others (Table 2). Interestingly, discussed later, many of these motifs are cognate sites for pioneer transcription factors involved in opening chromatin [32].

### 3.7. RSV-Induced Open Chromatin Peaks Are Associated with Active Promoters and JUN Binding

RSV-induced peaks also substantially overlapped with experimentally determined RNA Pol II and H3K27Ac peaks, indicating these sequences were enriched in activated promoters. Interestingly a substantial overlap of RSV-inducible nucleosome free regions was seen with JUN ChIP-Seq (Figure 7).

### 3.8. RSV-Induced Chromatin Domains Are Devoid of Tissue-Specific Enhancers

Although the majority of RSV-inducible chromatin regions mapped to intronic sequences (Figure 6A), we examined whether these sequences contained functionally active enhancers from A549 airway epithelial cells [31]. No overlap was seen between the two data sets (Figure 8). Collectively, we interpret these data to indicate that the RSV-induced open chromatin domains correspond primarily to introns in actively transcribed genes.

### 3.9. RSV-Induced Chromatin Opening Is Associated with Distinct Profiles of Gene Expression

We next sought to obtain insight into the relationship of RSV-inducible chromatin opening and gene expression. In this approach, RSV-induced ATAC-Seq peaks proximally located to TSSs were selected. These peaks were ±3000 bp of the TSS; of these, two-thirds were within <1 kB of the TSS, indicating that these were within the proximal promoter. We then quantified mRNA expression patterns for these genes by next generation sequencing. Short-read RNA-seq data from 0, 16, and 24 h of RSV-infected hSAECs were normalized, scaled, and subjected to hierarchical clustering (*n* = 4 independent samples at each time point). From the hierarchical clustering, we identified four distinct clusters of gene expression (Figure 9). These clusters included a cluster of genes that are induced after 24 h of infection; a cluster of genes induced at 16 h and fall by 24 h, and two clusters of genes whose expression declined over the course of infection (Figure 9).

To determine relationship with changes in chromatin accessibility and gene expression, the fold change in RSV-induced ATAC-Seq peak intensity of the nearest peak to the TSS was extracted and used to annotate each row of the hierarchical cluster. We found that genes with an increase in RSV-induced chromatin accessibility were more likely to show inductions of gene expression during the 24 h RSV infection, and genes whose expression fell were more likely to have reduced levels of chromatin accessibility (Figure 9). These findings suggest that promoters that have increased RSV-induced accessibility show enhanced gene expression.

### 3.10. Functional Enrichment of Gene Expression Changes

RSV-induced chromatin changes were analyzed by pathway enrichment, where signaling by receptor tyrosine kinases and ECM organization were the most enriched pathways (Figure 10A). Within these clusters, we noted that the genes encoding Glutamine-Fructose-6-Phosphate Transaminase 2 (*GFPT2*)*,* transforming growth factor B *(TGFB1), JUNB*, fibronectin (*FN1*), and matrix metalloprotein (*MMP9*) were identified. These genes play coordinate role in ECM remodeling in inducible epithelial mesenchymal transition [33,34].

An integrated network pathway analysis also identified three major hubs of ECM organization and adherent junctions interactions. Within these hubs, integrins, TGFβ, and cytoskeletal proteins (collagen and Laminin) are also identified (Figure 10B). Collectively, these functional analyses indicate that RSV-induced chromatin accessibility is associated with expression of TGFβ-induced ECM modifying proteins.

### 3.11. Growth Factor Induction of TGFB1-JUNB, FN1, and MMP9

To understand the effects of RSV infection on chromatin accessibility for the TGFβ-ECM pathway, an integrated genome viewer (IGV) visualization was performed. Here, counts of ATAC-Seq cleavage were mapped to individual gene features. In control hSAECs, *GFPT2* is in a partially open chromatin configuration, with peaks identified corresponding to the TSS and intron 5 (Figure 11A). The accessibility of *GFPT2* is increased upon RSV infection. The highly inducible *GFPT2* mRNA expression is also noted (Figure 11B).

Similarly, accessibility of the *TGFB1* TSS is substantially enhanced by RSV infection; this chromatin opening is associated with a fourfold increase in mRNA expression (Figure 11A,B). The nuclease cleavage pattern of *JUNB* is the most dramatic of the focus genes in the growth factor-ECM pathway. Here the proximal promoter, gene body, and 3′UTR of *JUNB* are dramatically increased in response to RSV infection. The proximal promoter of *FN1* is opened as a result of RSV infection; this gene shows a late pattern of expression (Figure 11B). Finally, RSV induces open chromatin domain of *MMP9,* both in its proximal promoter and a further upstream site, previously shown to be functionally important in *MMP9* expression.

## 4. Discussion

RSV is an important human pathogen responsible for acute inflammation and lower respiratory tract infection (LRTI). The mechanisms underlying chronic sequelae of RSV infection, including airway remodeling, atopy, and chronic reduction in pulmonary function are not fully understood. In experimental viral challenges and fatal cases of RSV LRTI, RSV replication is predominately seen in the airway epithelial cell [12,35]. Using ex vivo models of lower airway epithelial cells, we know that RSV infection is associated with global reprogramming of gene expression and protein secretion patterns. In this study, we extend the understanding of the effects of virus in this pathobiologically relevant model by focusing on understanding open chromatin states and inducible changes. High-confidence Tn5 transposase-accessible regions were predicted using two independent peak-calling algorithms. Here, we discover that open chromatin domains in resting lower respiratory cells are highly enriched in active promoters controlling housekeeping functions (organelle biosynthesis, protein *N*-glycosylation, and cell cycle regulation). By contrast, RSV-induced open chromatin domains are highly enriched in AP1 binding motifs and overlap JUN peaks in GEO ChIP-Seq data sets. Importantly, genes undergoing RSV induced remodeling are involved in growth factor-tyrosine signaling and ECM production. These data provide important insights into the mechanisms of RSV-induced structural remodeling that may be linked to chronic infectious sequelae.

Previous systems-level studies using gene profiling experiments have suggested that lower airway epithelial cells produce greater amounts of Th2-activating CCL-type chemokines than do epithelial cells of the conducting airways [10,36]. These findings were extended using unbiased comparative proteomics analysis. Here, proteins preferentially secreted by RSV infected bronchiolar cells compared to tracheal derived cells indicated enhanced secretion of Th2 activating cytokines (MIP1α, TSLP), mucin expressing (CCL20), and fibrogenic cytokines (IL6) [11]. Because RSV infection similarly activates NFκB, STAT, and IRF pathways in both upper and lower airway epithelial cells, the mechanisms for how preferential Th2 cytokine expression is exhibited by lower epithelial cells have not been determined. Here we provide direct evidence that these genes are contained in open chromatin domain in resting hSAECs. Substantial Tn5 transposase cleavage is seen on the IL6, TSLP, CCL20, and MIP1α promoters, and these promoters overlap experimentally determined H3K27Ac and RNA Pol II peaks. H3K27 marks are characteristic of inducible promoters bound by the positive transcriptional elongation factor [37]. With promoters in an open chromatin environment engaged with RNA Pol II, activated NFκB, STAT, and IRF are able to induce high level of expression in response to RSV infection. Together, these data indicate that promoters expressing this subgroup of Th2 cytokines are primed for rapid transcriptional elongation, a major mode of cytokine activation in response to the innate immune response [13,28].

We observe that RSV infection results in formation of some ~1700 open chromatin domains across the genome distributed over all chromosomes, including both sex chromosomes. These domains are associated with distinct functional patterns involving tyrosine kinase signaling and ECM organization. *TGFB1* expression is increased in RSV disease, where it induces collagen synthesis and airway remodeling [38]. However, the mechanism for how *TGFB1* is activated by RSV infection in epithelial cells has not been fully explained. Here, we observe that the *TGFB1* promoter is remodeled into an open state by RSV, and this process is associated with enhanced mRNA expression.

We also note that RSV induces chromatin opening of *GFPT2*, the rate limiting enzyme in the hexosamine biosynthetic pathway (HBP). Our previous work has shown that TGFB1 induces epithelial cell state transition and GFPT2 expression. Activation of the HBP was demonstrated by intracellular accumulation of uridine diphosphate N-acetylglucosamine (UDP-GlcNAc) and GFPT2-dependent FN1 glycosylation [33]. UDP-GlcNAc is the obligate activator of *O*-linked *N*-acetylglucosamine transferase (OGT). GFPT2 enhances capacity for protein glycosylation and folding, relieving proteostasis produced by increased ECM production and viral protein secretion, making its expression a key regulatory step in the unfolded protein response [33]. Although activation of the HBP has not been previously described in RSV infection, metabolomics profiling studies of airway epithelial cells indicate that *GFPT2* is upregulated in response to infection with human metapneumovirus [39]. Through the actions of GFPT2 and enhanced glycosylation of insoluble ECM proteins, such as FN1, RSV infected airway cells are able to produce more extracellular matrix, providing structural remodeling of the lung in LRTI.

MMP9 is a zinc-metalloproteinase involved in ECM remodeling, where it cleaves FN, Col IV, laminin, growth factor receptors, and other substrates. Previous work has shown that *MMP9* gene expression is induced by RSV infection, where it represents the major gelatinolytic activity in cultured supernatants [40]. The expression of *MMP9* has been studied in detail; it is known that *MMP9* transcription is controlled by cooperative AP1 and NFκB binding sites occluded by a nucleosome array in unstimulated cells. This nucleosome is repositioned upon stimulation with phorbol myristic acid (PMA)n a manner dependent on extracellular signal-regulated kinase (ERK) signaling, resulting in increased acetylated histone H3 binding [41]. Our findings showing increased chromatin accessible sites in the AP1/NFκB binding region of the upstream promoter are consistent with the nucleosome eviction model for *MMP9* activation [38].

Our data strongly suggest that RSV-inducible open chromatin domains are highly enriched in AP-1 binding, supported by motif analysis, and overlap with experimentally determined JUN binding in ChIP-seq. AP1 has been demonstrated to be essential for chromatin accessibility of glucocorticoid receptor [42], and involved in chromatin opening during T cell activation [43]. In these latter studies, consistent with our findings here, inducible AP1 binding was observed in open chromatin produced by T cell receptor (TCR) stimulation. We note studies that show systemic induction of c-JUN results in organ fibrosis in mice [44]. RSV-induced AP1 activation is coordinated via post-translational modifications, including JUN N terminal kinase-mediated phosphorylation and upregulated mRNA expression of *JUNB* and *FOS* [45].

We interpret the findings that TGFβ-ECM pathway genes are regulated by enhanced chromatin accessibility to suggest the functional role of pioneer factors in RSV-induced gene reprogramming. Pioneer factors are a subclass of transcription factors that activate repressed genes important in cell state transition and differentiation. A number of pioneer factors have been identified, including FoxA, p53, and NFY [46]; we note binding motifs for this subclass of transcription factors are enriched in RSV-induced open chromatin environments (Table 2). Although most models of cellular reprogramming have been developed to study models of development and oncogenic transformation, we contend that cellular reprogramming also occurs in airway epithelial cells in response to virus infection and chronic inflammation (asthma and chronic obstructive pulmonary disease). For example, chronic activation of innate signaling via Toll like receptor 3 induces epithelial mesenchymal transition (EMT) through a complex series of partial cell state transitions mediated by RELA [26,34]. The role of AP1 as a pioneer factor in innate signaling-induced EMT has not been explored to our knowledge.

Activation of growth factor-TGFβ-HBP and extracellular remodeling may play an important role in structural remodeling of the airway, linking LRTI with chronic sequelae. Mentioned earlier, RSV replicates for up to 28 d in the lower respiratory tract in viral challenge models [12], where chronic inflammation and signals may be generated to promote disruption of the epithelial barrier and structural remodeling. Viral infection induced EMT induces secretion of IL6, TGFβ, and other growth factors that trigger expansion of subepithelial myofibroblasts [13]. Myofibroblasts are αSMA^+^/COL1^+^ and are thought to be major sources of ECM deposition, contributing to lamina reticularis expansion and interstitial fibrosis [47]. This process may be important in the reduction of lung function seen in observational studies of LRTI cohorts [48].

### 4.1. Limitations

In this first description of chromatin remodeling, we have chosen to use very stringent filter requiring chromatin domains to be independently predicted by two unrelated Peak Calling algorithms. This initial analysis will undoubtedly be expanded as more sensitive and reproducible informatics algorithms are developed. Our experimental model is a standardized RSV infection using a multiplicity of infection of 1. In this model, RSV replication activates the NFκB and IRF transcription factors in defined kinetics [28], producing innate gene expression with minimal effects on cell death [11]. The kinetics of gene expression, cell death and presumably chromatin remodeling will be influenced by changes in MOI. Our study is not designed to address whether similar or distinct effects of chromatin remodeling occur with other viral infections, but this is an interesting question to pursue.

### 4.2. Future Directions

In future studies, it will be of interest to test the role of JUN as a virus induced pioneer transcription factor and understand its role and mechanism in chromatin opening. It will be of interest to examine the effects of other respiratory viruses on chromatin regulation.

## Figures and Tables

**Figure 1 viruses-12-00804-f001:**
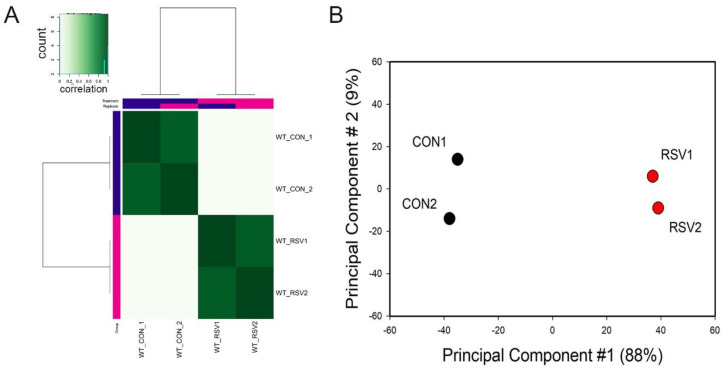
Sample QC. (**A**) Correlation analysis of the assay for transposase-accessible chromatin sequencing (ATAC-Seq) reads for the control and respiratory syncytial virus (RSV)-infected samples. Samples were subjected to hierarchical clustering, with distance indicated in the dendrogram. Note that the individual control (CON) and RSV samples cluster together and the two CON samples are distant from the RSV infected samples. (**B**) Principal Component Analysis. ATAC-Seq peaks were subjected to PCA. X axis, first principal component; Y axis, second principal component. Note the first principal component separates the RSV from the CON treated samples and represents 88% of the variability in the experiment.

**Figure 2 viruses-12-00804-f002:**
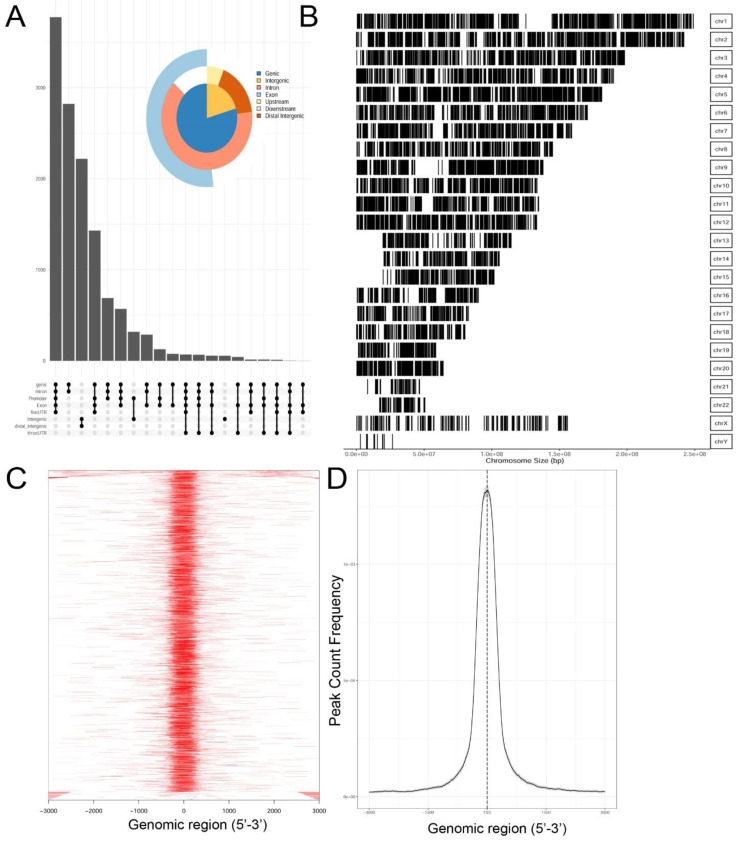
Occupancy analysis of control, human small airway epithelial cells (hSAECs). (**A**) “Upset” plot of genomic annotations of ATAC-Seq peaks. The upset plot provides information on correspondence of multiple peaks in a gene body. Note the most frequent pattern are peaks contain the promoter, 5′UTR, exon, intron, and 3′ UTR. Inset, pie chart of genomic region annotation. (**B**) Genome coverage of open chromatin domains. Shown is a linear representation of open peaks for each chromosome (labeled at right). (**C**) Tagged heat map of open chromatin domains relative to the 5′ regulatory region from −3000 nt to +3000 nt. Each red bar represents the length of open chromatin domain on a gene. (**D**) Distribution of peaks relative to the transcription start site (TSS). Note symmetric distribution over the TSS.

**Figure 3 viruses-12-00804-f003:**
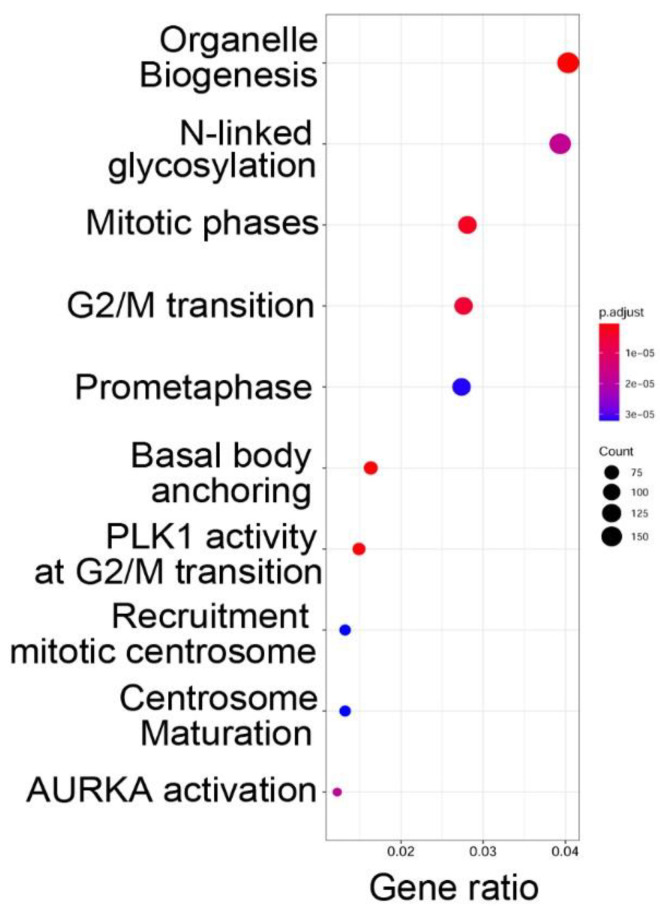
Gene pathways associated with open chromatin domains in hSAECs. Shown are genome ontologies ranked by the number of genes in a pathway (gene ratio) and by enrichment relative to genome (adjusted *p* value, padjust). N, NH2 terminal; G2/M, cell cycle phases; PLK1 Polo Like Kinase 1, AURKA, Aurora Kinase A.

**Figure 4 viruses-12-00804-f004:**
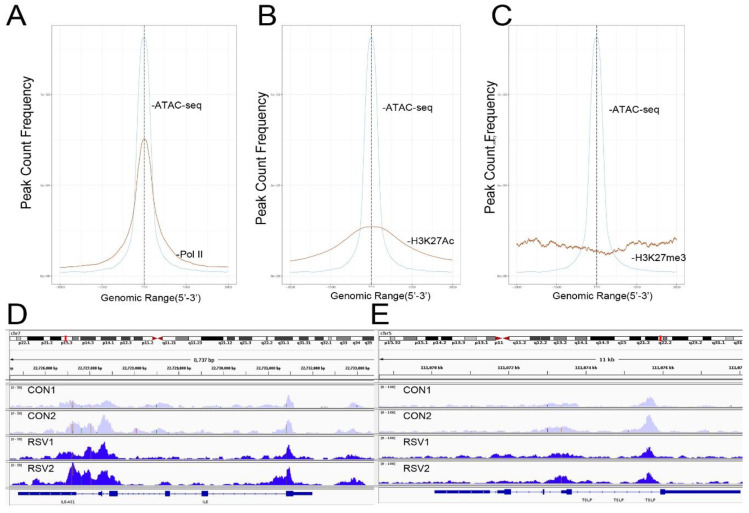
Open chromatin domains are enriched in active promoters. (**A**) Transcription factor overlap of ATAC-seq signals (blue tracing) with RNA Pol II (brown tracing). (**B**) Overlap with H3K27Ac. (**C**) Overlap with H3K27me3. (**D**) Integrated genomic viewer (IGV) of IL6 gene in control (purple) and RSV-infected state (blue). Note the substantial cleavage of IL6 gene body and proximal promoter in control cells, indicating chromatin accessibility. (**E**) IGV of TSLP. Transcription is from left to right. Transposase accessible regions cover the promoter, intron 2, and distal portion of intron 3.

**Figure 5 viruses-12-00804-f005:**
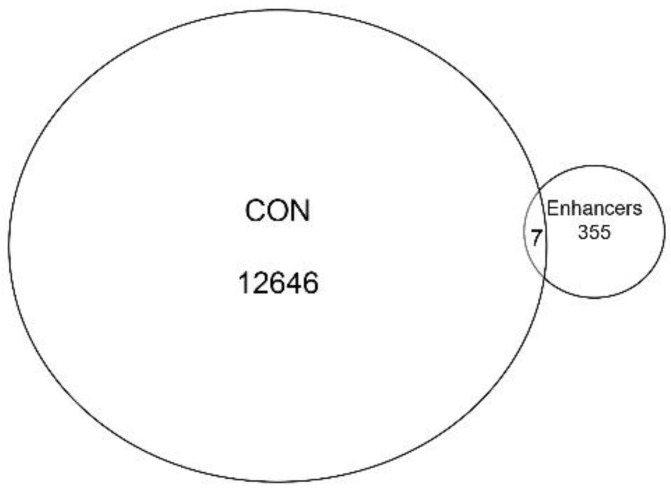
Presence of active enhancers in ATAC-Seq peaks. Venn diagram of the overlap between the high confidence ATAC-Seq peaks from uninfected hSAECs and active A549 enhancer sequences.

**Figure 6 viruses-12-00804-f006:**
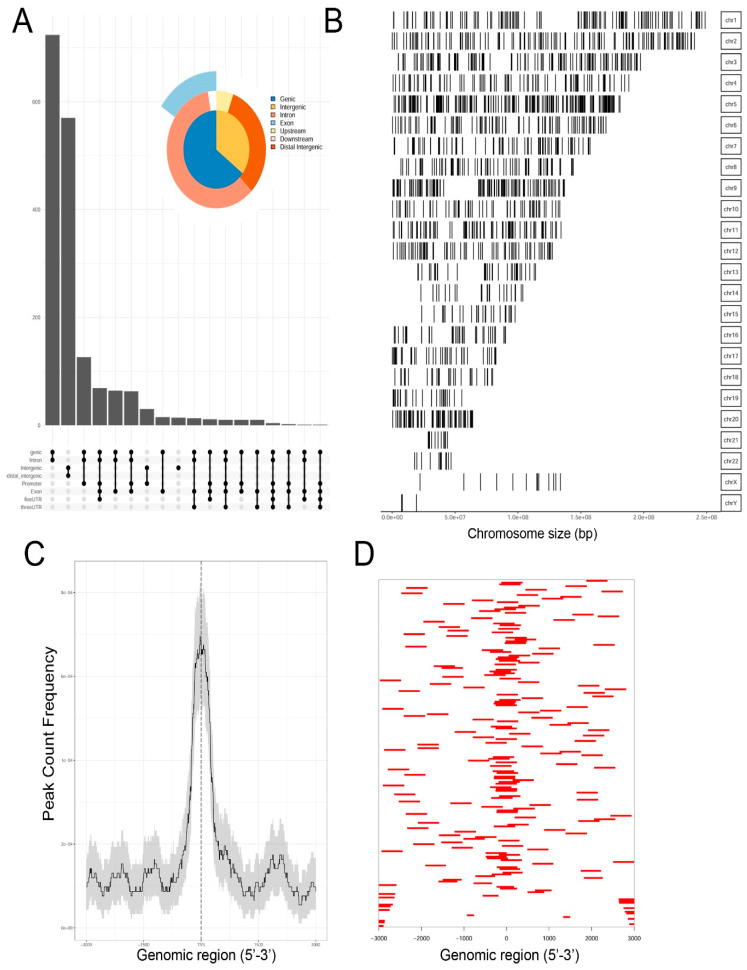
Analysis of RSV-induced open chromatin domains. (**A**) Upset plot of genomic annotations of ATAC-Seq peaks. Note the highest frequency of sites corresponds to intronic locations in gene bodies. (**B**) Genome coverage of open chromatin domains each chromosome (at right). (**C**) Distribution of peaks relative to the transcription start site (TSS). (**D**) tagged heat map of open chromatin domains relative to the 5′ regulatory region from −3000 nt to +3000 nt. Each red bar represents the length of open chromatin domain on a gene.

**Figure 7 viruses-12-00804-f007:**
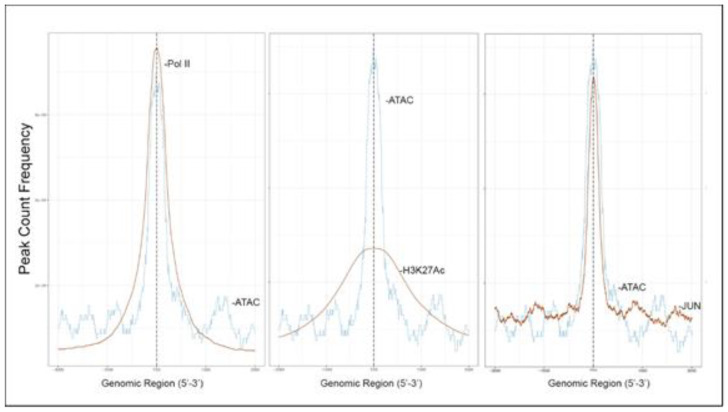
RSV-induced promoters are enriched in active histone marks and JUN/AP1 binding. Shown are transcription factor overlap plots of ATAC-seq signals with ChIP-seq binding of RNA Pol II, H3K27Ac, and JUN by Peak Count Frequency. Note the H3K27Ac binding is consistent with the broad domains produced by histone modification in contrast to the narrow peaks corresponding to polymerase and transcription factor binding.

**Figure 8 viruses-12-00804-f008:**
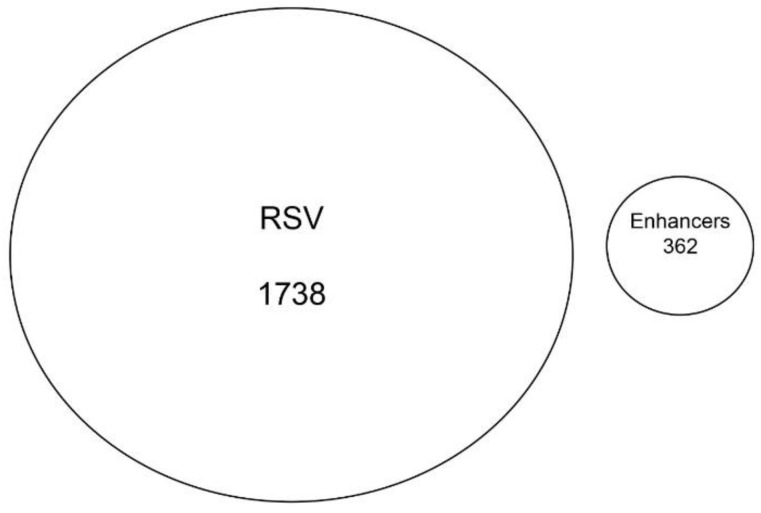
RSV-induced ATAC-Seq peaks are devoid of functional enhancers. Venn diagram of the overlap between the high confidence ATAC-Seq peaks from RSV-infected cells and active A549 enhancer sequences.

**Figure 9 viruses-12-00804-f009:**
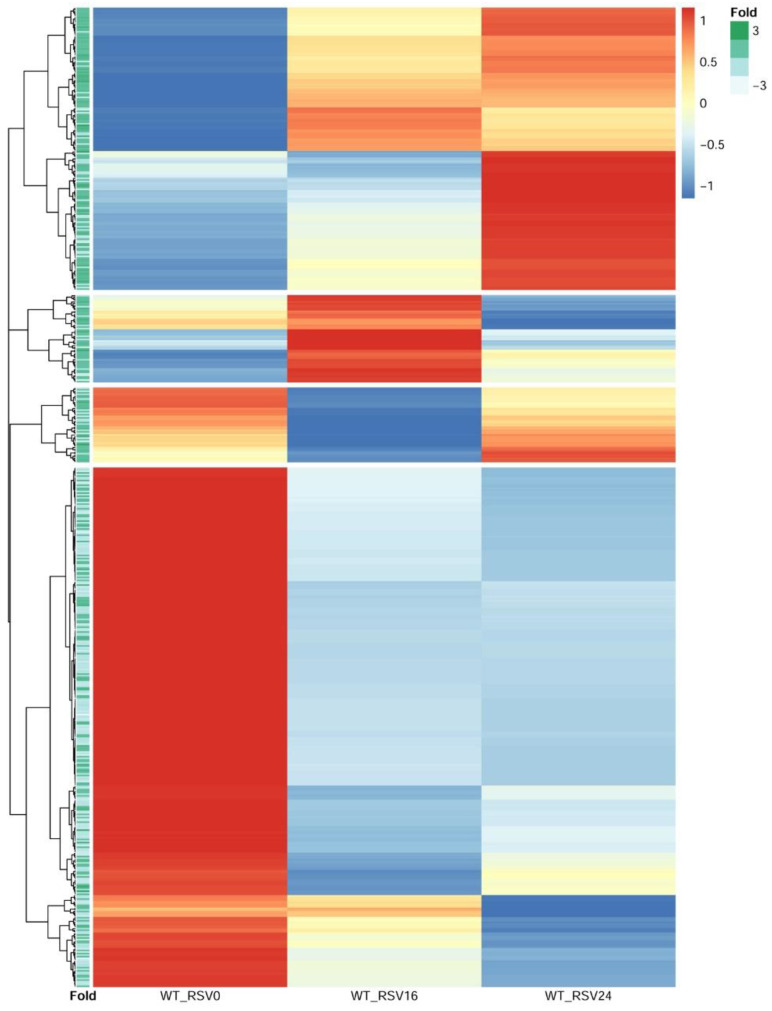
mRNA expression changes associated with RSV-induced changes in chromatin accessibility. mRNA expression patterns of genes within 3000 bp of RSV induced nucleosome free regions were scaled and grouped by hierarchical clustering using Euclidian distance as the distance metric. Each row represents a different gene. Four distinct expression patterns are separated by white space. Bottom left, the “Fold” annotation column indicates the fold change in accessibility of the nearest ATAC-seq peak to the TSS. Note the higher accessibility scores tend to occur in the top 2 clusters; expression of these genes is increased by RSV infection.

**Figure 10 viruses-12-00804-f010:**
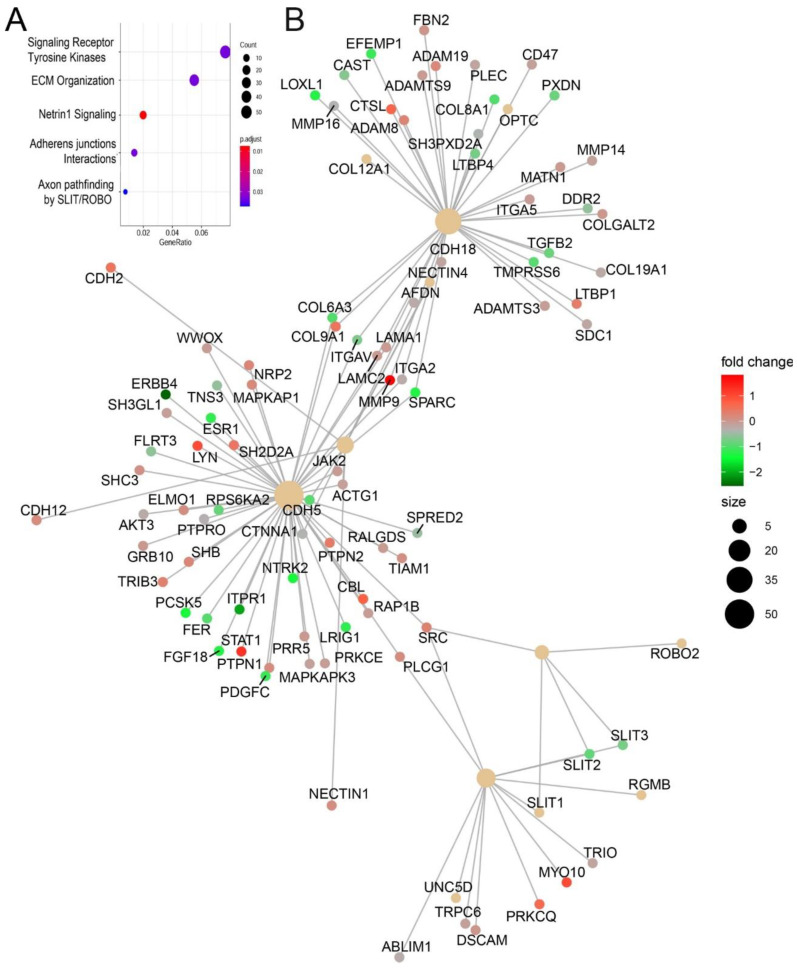
Pathway analysis of genic regions associated with RSV induced open chromatin domains. (**A**) Shown are genome ontologies ranked by the number of genes in a pathway (gene ratio) and by enrichment relative to genome (*p* value). (**B**) Integrated network analysis of genes shows 3 major functional pathways are affected by RSV induced changes in chromatin accessibility. Node size and fold change in nucleosome free domain are indicated in the legend.

**Figure 11 viruses-12-00804-f011:**
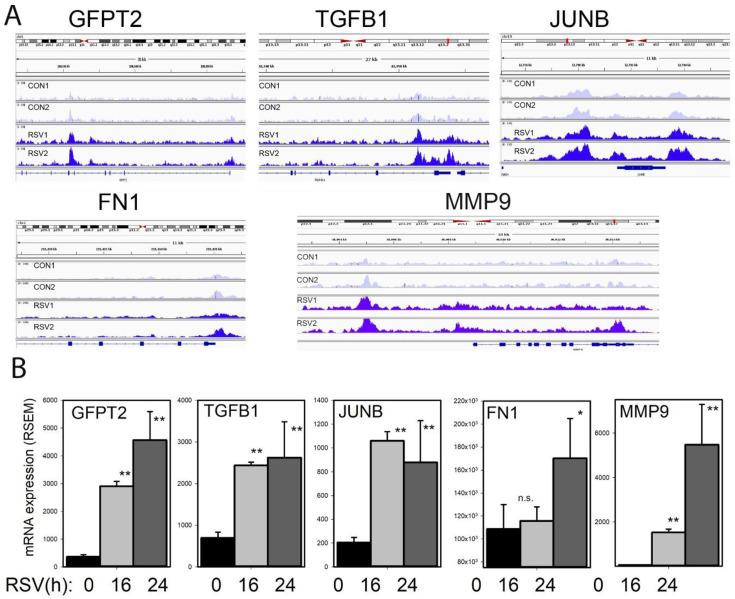
Inducible chromatin changes in the TGFβ growth factor-ECM pathway. (**A**) Integrated genomic viewer (IGV) of ATAC-Seq cleavage fragments mapped to the *GFPT2, TGFB1, JUNB,* and *FN1* genes in control and RSV-infected state. Note the increased transposase digestion of promoter elements after RSV infection. (**B**) mRNA changes of each gene. Shown is mean ± SD (*n* = 4 independent RNA-seq reads quantified by RSEM). *, *p* < 0.05, **, *p* < 0.01 post hoc Tukey’s test.

**Table 1 viruses-12-00804-t001:** Motif analysis.

Sequence	Name/Class (Related Sequences)	*p*-Value	%Target Sequences
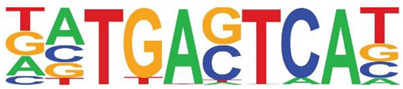	ATF/BZIP(Fra, Jun, Fos, AP-1, JunB, Fosl2, Bach2)	1 × 10^−1203^	15%
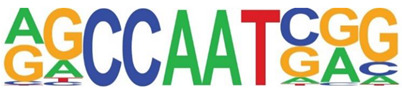	NFY/CCAAT	1 × 10^−317^	9.9
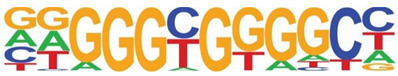	KLF1(Zf)(SP1,SP2, KLF5, KLF3, KLF6,KLF14)	1 × 10^−290^	15.7
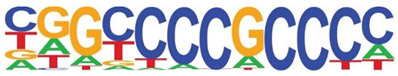	SP2/ZF(SP1, SP5)	1 × 10^−285^	22
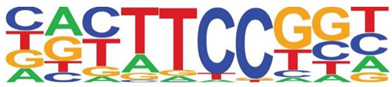	Fli1/ETX	1 × 10^−170^	11
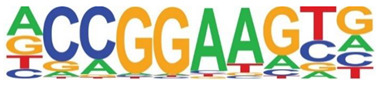	ETV4/ETS	1 × 10^−139^	11
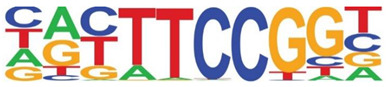	ELK4/ETS	1 × 10^−134^	7
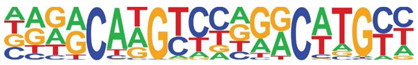	P73/p53(p53)	1 × 10^−132^	1.3
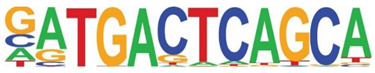	NF-E2/BZIP	1 × 10^−114^	1.5

BZIP, basic domain-leucine zipper; AP1, activator protein 1; ETS, E26 transformation-specific; NFY, nuclear transcription factor Y; KLF, Kruppel Zinc Finger; and SP1, specificity protein 1.

**Table 2 viruses-12-00804-t002:** Motif analysis of RSV induced nucleosome free regions.

Sequence	Name/Class (Related Sequences)	*p*-Value	%Target Sequences
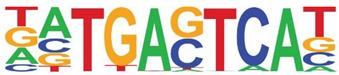	ATF/BZIP (Fra, Jun, Fos, AP-1, JunB, Fosl2, Bach2)	1 × 10^−187^	19%
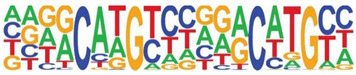	P63/p53 (p73,p53)	1 × 10^−39^	6.5
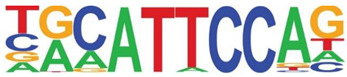	TEAD3/TEA	1 × 10^−28^	10.4
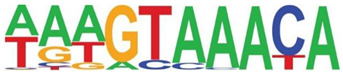	FOXA1/Forkhead	1 × 10^−22^	8.4
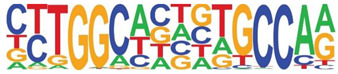	NF1/CTF	1 × 10^−21^	4.8
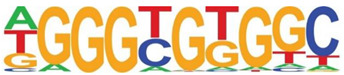	KLF5	1 × 10^−21^	4.8
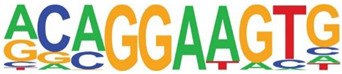	ERG/ETS	1 × 10^−18^	8.3
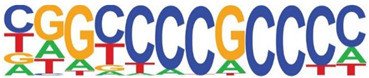	SP2	1 × 10^−14^	10.3
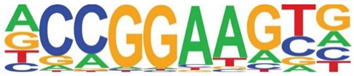	ETV4/ETS	1 × 10^−14^	8.1

Abbreviations used are: BZIP, basic domain-leucine zipper; AP1, activator protein 1; ETS, E26 transformation-specific; NFY, nuclear transcription factor Y; KLF, Kruppel Zinc Finger.

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
