# Peer review of "Respiratory Syncytial Virus Infection Induces Chromatin Remodeling to Activate Growth Factor and Extracellular Matrix Secretion Pathways"

_viruses, 2020, doi:10.3390/v12080804_

Round 1

Reviewer 1 Report

Xu et al. use a common combination of ATAC- and RNA-seq to study the impact of RSV infection on the chromatin landscape. I am not an expert in virology, thus will only comment on the eppigenomic analysis. The data seems to be fine and the quality metrics for the ATAC-seq are within the range of what has been reported in other studies. One remark though is that the periodicity of the nucleosomal fragments (insert Supplementary Figure 1) is not ideal and the individual fragments should be much easier to distinguish. Thus, I would like to see a correlation analysis (better even a PCA) to be able to judge reproducibility of the sequencing data.
Although present at active promoters, H3K27ac is also a classical active enhancer mark - to complete the analysis it would be nice to consider enhancer activity (particularly with respect to TF binding) and overlap the ATAC-seq signal and k27ac outside promoter regions. Subsequently, openess of enhancers should be correlated with transcriptional activity.
Clearly, the results section needs extensive editing - the manuscript is very difficult to follow and very confusing.

Author Response

Reviewer Response

Xu et al.

We thank the reviewers for their comments and suggestions for improving the manuscript.  We have responded to each comment through additional analysis, modifying the text and introducing new Figures (1, 5, 8). Changes are not marked in the manuscript, but detailed below.

Reviewer 1

Comment #1:  I would like to see a correlation analysis and PCA to evaluate the reproducibility of the data. 

Response:  We include this information as Figure 1 in the revised manuscript.  In Figure 1A, the correlation plot indicates that the two replicates from control and RSV infected cells are grouped together and quite distinct based on the treatment.  In Figure 1B,  a PCA plot indicates that the ATAC-Seq peaks from the control cells and RSV-infected cells are widely separated.  Note that over 88% of the experimental variability is contained in the first principal component, separating the treatment conditions, whereas 9% of the variability is due to sample replicates. Together these data indicate that the replicates from each treatment are similar and capture the effect of RSV infection, strengthening the robustness of our data.

Comment #2:  To complete the analysis, it would be nice to consider enhancer activity, since H3K27Ac is also seen in enhancers.

 Response:  Thank you for this suggestion.  Having said that, annotation of enhancer sequences is not as straightforward as annotation of transcription start sites, a problem compounded by the knowledge that the location of enhancers is cell-type (and probably stimulus dependent).  As noted by the reviewer, histone marks can be located in both promoters, active gene bodies and enhancers. To approach this problem more rigorously, recent studies have indicated enhancer sequences can be identified by the presence of bidirectional enhancer RNA (eRNAs) in genome run-on sequencing (Gro-Seq).  We were able to identify functional enhancers from closely related human alveolar airway cells, identified by bidirectional eRNA expression in the annotated HACER database [1].  Examining the overlap of constitutively open chromatin domains and functional airway epithelial cell enhancers, we found only 7 enhancers in the constitutively open chromatin domains (Figure 5 of the revised manuscript).  Motif analysis of these 7 peaks and/or genes associated with them (<0.05% of the data set) will not be representative of the constitutive ATAC-Seq and was not further pursued. 

We therefore state in the Results section that “Collectively, the results of the gene annotation (Figure 2A), tagged matrix (Figure 2C), enrichment of peaks to annotated TSSs (Figure 2D), and co-occurrence with RNA Pol II ChIP-Seq peaks lead us to conclude the ATAC-Seq cleavage sites are enriched in proximal promoter sequences.”  

A similar analysis was conducted for the RSV-induced ATAC-Seq peaks.  No overlap with the functionally defined enhancer sequences was detected (Figure 8).  The significant enrichment of intronic sequences suggests that the RSV-inducible ATAC-seq sequences are located within gene bodies and not enhancers, also stated in the Results of the revised manuscript.

Comment #3:  the manuscript is difficult to follow..

Response:  We apologize for the terse writing style.  We have extensively reworked the Results section to make the analyses and figures more accessible for the audience.

Reviewer 2 Report

Xu et al performed an extensive and comprehensive in vitro study to assess possible mechanisms surrounding the decline in lung functions after RSV infection.

Results are densely written and sometimes hard to follow. Given the importance of this preliminary study to answer such an important questions I encourage authors to review this section, since I fear they will lose a good amount of readers that might not be as familiar with the techniques applied and results showed. 

1) Please include as supplementary figure the multidimensional plots showing co-clustering from each treatment.

2) Authors mentioned Figure 2 C, D but Figure 2 only has one panel. Please clarify since I think they mean Figure 1 C, D. (line 166)

3) Samples were obtained at 0, 16 and 24 hs. However, I do not see any comparison between timelines. How different are this samples and do they indicate any sort or progression? Is there any value or consideration in obtaining samples at 48 or 72 hs? A progression in time of the results would be interesting.

4) Authors should include in their discussion whether time of sample collection and / or MOI used could play a role in the results.

5) Is there anything to make authors think this is specific to RSV? We know that after bone marrow transplant infection with RSV, HMPV or PIV (especially the latter) can induce long term decline in pulmonary function and lung restructuration/fibrosis.

Author Response

Reviewer 2

Comment #1:  The authors are encouraged to review the Results section to improve readability.

Response:  Thank you for this feedback.  We have extensively revised the results section to more clearly explain the informatics and experimental interpretation. 

Comment #2:  Please include the multidimensional plots showing co-clutering of the data.

Response:  Please see comment 1 to Reviewer 1, above, and Figure 1 of the revised manuscript.

Comment #3:  Please clarify the figure citation on line 166; is this Figure 1?.

Response:  The reviewer is correct; we apologize for the typographical error.  The figure is now correctly cited in the paragraph beginning on line 205.  The revised Figure is now 2.

Comment #4:  Samples are obtained at 16 and 24 h, how different are these?

Response:  Changes in chromatin remodeling is the focus of the paper.  The ATAC-Seq sampling is limited to only 0 and 24 h time points.  The correlation of RNA-seq data included an intermediate 16 h time point, a time before maximal activation of innate response.  These changes in RNA expression have been extensively reported previously [2-12].  Data sampling after 24 h is difficult because of substantial cell death.

Comment #5:  Authors should note whether time of collection and MOI could play a role in the results.

Response:  We describe in the “Limitations” section that “Our experimental model is a standardized RSV infection using a multiplicity of infection of 1. In this model, RSV replication activates the NFkB and IRF transcription factors with defined kinetics [28], producing innate gene expression with minimal effects on cell death [11].  Higher MOIs induced more rapid gene responses and earlier cell death.  The kinetics of chromatin remodeling will undoubtedly be influenced by changes in MOI.  Exploration of the effects of MOIs is beyond the scope of this study. 

Comment #6:  Is there anything to make the authors think this is specific to RSV?

Response:  The expression of nonstructural viral proteins influences the genomic response.  For example, we have observed that distinct genomic responses occur in hSAECs infected with two closely related paramyxoviruses, hMPV and RSV.  However, our study is not designed to address whether similar or distinct effects of chromatin remodeling occur with other viral infections, but is an interesting question to pursue.  This is stated in the “Limitations” and “Future Directions”.

References for Reviewer Response

  1. Wang, J., X. Dai, L. D. Berry, J. D. Cogan, Q. Liu and Y. Shyr. "Hacer: An atlas of human active enhancers to interpret regulatory variants." Nucleic Acids Res 47 (2019): D106-D12. 10.1093/nar/gky864. https://www.ncbi.nlm.nih.gov/pubmed/30247654.
  2. Zhang, Y., B. A. Luxon, A. Casola, R. P. Garofalo, M. Jamaluddin and A. R. Brasier. "Expression of respiratory syncytial virus-induced chemokine gene networks in lower airway epithelial cells revealed by cdna microarrays." J Virol 75 (2001): 9044-58. 10.1128/JVI.75.19.9044-9058.2001. http://www.ncbi.nlm.nih.gov/pubmed/11533168.
  3. Tian, B., Y. Zhang, B. A. Luxon, R. P. Garofalo, A. Casola, M. Sinha and A. R. Brasier. "Identification of nf-kappab-dependent gene networks in respiratory syncytial virus-infected cells." J Virol 76 (2002): 6800-14. http://www.ncbi.nlm.nih.gov/pubmed/12050393.
  4. Zhang, Y., M. Jamaluddin, S. Wang, B. Tian, R. P. Garofalo, A. Casola and A. R. Brasier. "Ribavirin treatment up-regulates antiviral gene expression via the interferon-stimulated response element in respiratory syncytial virus-infected epithelial cells." J Virol 77 (2003): 5933-47. http://www.ncbi.nlm.nih.gov/pubmed/12719586.
  5. Choudhary, S., S. Boldogh, R. Garofalo, M. Jamaluddin and A. R. Brasier. "Respiratory syncytial virus influences nf-kappab-dependent gene expression through a novel pathway involving map3k14/nik expression and nuclear complex formation with nf-kappab2." J Virol 79 (2005): 8948-59. 10.1128/JVI.79.14.8948-8959.2005. http://www.ncbi.nlm.nih.gov/pubmed/15994789.
  6. Liu, P., K. Li, R. P. Garofalo and A. R. Brasier. "Respiratory syncytial virus induces rela release from cytoplasmic 100-kda nf-kappa b2 complexes via a novel retinoic acid-inducible gene-i{middle dot}nf- kappa b-inducing kinase signaling pathway." J Biol Chem 283 (2008): 23169-78. 10.1074/jbc.M802729200. http://www.ncbi.nlm.nih.gov/pubmed/18550535.
  7. Liu, P., M. Lu, B. Tian, K. Li, R. P. Garofalo, D. Prusak, T. G. Wood and A. R. Brasier. "Expression of an ikkgamma splice variant determines irf3 and canonical nf-kappab pathway utilization in ssrna virus infection." PLoS ONE 4 (2009): e8079. 10.1371/journal.pone.0008079. http://www.ncbi.nlm.nih.gov/pubmed/19956647.
  8. Brasier, A. R., B. Tian, M. Jamaluddin, M. K. Kalita, R. P. Garofalo and M. Lu. "Rela ser276 phosphorylation-coupled lys310 acetylation controls transcriptional elongation of inflammatory cytokines in respiratory syncytial virus infection." J Virol 85 (2011): 11752-69. 10.1128/JVI.05360-11. http://www.ncbi.nlm.nih.gov/pubmed/21900162.
  9. Tian, B., Y. Zhao, M. Kalita, C. B. Edeh, S. Paessler, A. Casola, M. N. Teng, R. P. Garofalo and A. R. Brasier. "Cdk9-dependent transcriptional elongation in the innate interferon-stimulated gene response to respiratory syncytial virus infection in airway epithelial cells." J Virol 87 (2013): 7075-92. 10.1128/JVI.03399-12. http://www.ncbi.nlm.nih.gov/pubmed/23596302.
  10. Fang, L., S. Choudhary, B. Tian, I. Boldogh, C. Yang, T. Ivanciuc, Y. Ma, R. P. Garofalo and A. R. Brasier. "Ataxia telangiectasia mutated kinase mediates nf-kappab serine 276 phosphorylation and interferon expression via the irf7-rig-i amplification loop in paramyxovirus infection." J Virol 89 (2015): 2628-42. 10.1128/JVI.02458-14. http://www.ncbi.nlm.nih.gov/pubmed/25520509.
  11. Zhao, Y., M. Jamaluddin, Y. Zhang, H. Sun, T. Ivanciuc, R. P. Garofalo and A. R. Brasier. "Systematic analysis of cell-type differences in the epithelial secretome reveals insights into the pathogenesis of respiratory syncytial virus-induced lower respiratory tract infections." J Immunol 198 (2017): 3345-64. 10.4049/jimmunol.1601291. https://www.ncbi.nlm.nih.gov/pubmed/28258195.
  12. Tian, B., J. Yang, Y. Zhao, T. Ivanciuc, H. Sun, M. Wakamiya, R. P. Garofalo and A. R. Brasier. "Central role of the nf-kappab pathway in the scgb1a1-expressing epithelium in mediating respiratory syncytial virus-induced airway inflammation." J Virol 92 (2018): 10.1128/JVI.00441-18. https://www.ncbi.nlm.nih.gov/pubmed/29593031.

Round 2

Reviewer 1 Report

The authors have addressed all my concerns and I am happy to suggest publication. I would leave it to the authors to remove the enhancer analysis (especially the Venn diagram) as it does not add much. Maybe it is worth mentioning in the text without an additional figure.